# Parents of Children with Congenital Heart Disease (CHD): A Narrative Study of the Social and Clinical Impact of CHD Diagnosis on Their Role and Health

**DOI:** 10.3390/bs15030269

**Published:** 2025-02-25

**Authors:** Christian Moro, Antonio Iudici, Gian Piero Turchi

**Affiliations:** Department of Philosophy, Sociology, Education and Applied Psychology, University of Padova, 35131 Padova, Italy; christian.moro@unipd.it (C.M.); antonio.iudici@unipd.it (A.I.)

**Keywords:** congenital heart disease, chronic illness, parents, psychological impact, social impact, narratives, discourse analysis, text analysis, MADIT, health

## Abstract

Congenital heart diseases (CHDs) lead to psychological and social repercussions for parents of affected children: the diagnosis, screenings, surgeries, and hospitalization, as well as ongoing difficulties bring with them stress, anxiety, fear, stigmatization, and isolation. Studies investigating parents’ direct perspective on these issues lack in the field literature. Our research aims to leverage parents’ narratives in order to explore how they describe their role as parents of a child with CHD and the impact of its social and clinical repercussions on their lives. We recruited 45 parents and analyzed their narratives through the MADIT approach, focusing on the discursive modalities and content cores employed. Parents describe and judge their role as ‘worried-protective’, ‘heroic’, ‘normal-untroubled’, and ‘unfortunate’, in a way that strongly characterizes the person, leaving limited possibilities for assuming different features. The clusters ‘state of ordeal’, ‘state of alert-overprotection’, and ‘personal identity changes’ are connoted as inevitable and established component of parents’ lives, while ‘limitation of life experiences’ is less monolithic and more open to change. Current narratives assume a totalizing form in the life of these parents, that can lead to stigma and exacerbate the already present difficulties and challenges, that need targeted psychological intervention by field professionals.

## 1. Introduction

Congenital heart disease (CHD) is defined as a problem in the structure of the heart, already present at birth, that may affect the shape and overall functioning of the heart or some of its parts ([42]). This can lead to problems in body function and development, as well as general health. Congenital heart defects are currently the most common type of defects present at birth ([32]), with an occurrence in the global population of approximately 1% ([55]). Moreover, they are the main cause of disability and/or mortality in childhood ([40]). To date, thanks to improvements in the care and treatment of these defects, congenital heart disease has gone from being a terminal to a chronic condition, and over 85% of newborns with CHD, even complex, survive after the first year of life and into adulthood ([39]; [43]).

To date, when an infant is diagnosed with CHD, a medical-surgical plan is initiated, and for the parents of the child this entails ongoing challenges ([64]). Parents face doctor’s appointments, screenings, potentially fatal interventions such as open-heart surgery ([20]), prolonged periods of hospitalization and care, multiple surgeries, near-death experiences, and, in the most unfortunate cases, death ([8]; [22]; [34]; [52]). It is reported that the child’s CHD is a source of stress for the whole family ([21]; [65]); reviews and quantitative and qualitative surveys indicate how parents can experience anxiety, depression, somatization ([3]; [5]; [67]), stress, loss of hope, and social isolation. Such psychological and social sequelae are associated with the diagnosis of CHD, its discovery and acceptance ([5]; [35]; [48]), the period of the child’s hospitalization, the medical/hospital context ([15]; [21]; [45]), and the perception of disease and parenting of a child with chronic heart disease ([35]; [54]).

Moreover, regarding this latter aspect, social stereotyping can influence both children’s and parents’ coping behaviors, exacerbating their social isolation and marginalization ([12]). For example, following the notice of their child’s diagnosis, some parents start isolate themselves from others and avoid social interactions ([41]). Similarly to parents of children with other chronic diseases such as diabetes and thalassaemia, parents of children with CHD also experience stigmatization and social constraints ([12]; [27]; [38]). They are faced with other’s negative perceptions or beliefs about them and also worry that they and their child may be discriminated ([68]). This could also lead, in addition, to both children with CHD and their parents running into disclosure dilemmas related to revealing the medical condition ([37]); in fact, due to social stigma and the misjudgment of others, both parents and children feel insecure about who disclose the congenital heart disease to ([68]). Therefore, fearing social insensitivity, they tend to keep the situation and the burden it carries secret ([37]). Some parents also tend to keep their emotions hidden, due to external pressure ([16]). Indeed, they feel the expectation to be strong and ‘hold everything inside’, as well as the potential judgment of other to not be able to properly care for their child ([16]). Last but not least, the economic dimension that affect the parents’ overall health needs to be considered; in fact, they experience greater financial instability due to sick leave and medical expenses sustained to care for their children with CHD, factors that lead to a reduction in their standard of living ([30]; [64]). Thus, the burden of care is placed squarely on the parents, who as primary caregivers bear duress and consequences throughout the entire period of dealing with their child’s CHD ([29]; [57]; [64]). This care burden, combined with other detrimental factors linked to the chronic illness (e.g., social isolation), can reduce familial wellbeing and quality of life ([28]).

Therefore, it is crucial to consider the care and involvement of caregivers as paramount, as it impacts not only on their lives, but also—and especially—on those of their children with CHD. In fact, depression, anxiety, stress, and hopelessness can affect both parental wellbeing and the child’s treatment, where this latter—if successful—is a positive factor that can improve parents’ hopefulness ([33]).

As seen above, the currently available literature shows many quantitative and qualitative studies concerning the psychological and social repercussions that afflict parents of children with CHD. However, there are only few studies on how parents interact with their children with CHD and how they specifically narrate their experience and their role. Starting from this, we defined the following research questions: do all aspects of the parent’s daily life revolve around the illness and are, therefore, permeated by it, limiting the possibility of creating new narratives? Which narrative modalities do parents of a child with CHD use to tell their story about the disease, the medical symptoms on their child and the psychosocial repercussions on the whole family?

Considering this gap in the studies on the subject, we have set up a dedicated study aimed at (1) describing how the parents of a child with CHD narratively configure their role and (2) describing how these parents recount the impact of CHD’s psychological and social repercussion on their lives. We performed textual analysis on the narratives of parents affiliated with an Italian CHD association, to be able to further and better qualify their experience. The end goal was to provide data useful to improve the indirect care toward people with CHD, precisely by leveraging parents’ narratives to enhance—through dedicated interventions—the effectiveness of their actual and potential support (for themselves, other parents, and their children).

## 2. Methods

### 2.1. Methodological Assumptions

The reference framework of our work lays in the interactionist and narrativistic paradigm, which assumes that the way reality is constructed is in interaction ([10]; [19]; [36]; [47]; [50], [51]). The conceptual core of this perspective argues that the process through which people construct their identity depends on the way they interact ([58]).

Consistently with this assumption, textual data analysis has always been one of the most widely used tools in research ([56]; [7]; [11]; [13]; [14]; [46]). Interest in the systematic analysis of texts has increased significantly in recent decades ([25]). Within this methodological tradition, in our research we adopted the Methodology for the Analysis of Computerized Text Data (MADIT) ([2]; [23]; [59], [60]). The MADIT pertains to discourse analysis methods and focuses on identifying the narrative and discursive processes of participants’ texts. These processes have been codified through a specific definition (Discursive Repertory—see below), from which the analysis of peoples’ narratives is articulated. The aim of the MADIT is to offer a picture of how people’s narratives generate a reality of sense, not only in ‘verbal’ terms but also in terms of pragmatic fallouts of that reality itself. In other words, it offers a snapshot of how the narrative related to living in close proximity with CHD is oriented towards scenarios in which individuals can say they are “in health” ([62]).

MADIT has been applied in different areas, such as clinical psychology ([2]), disability ([26]), immunodeficiencies ([24]), social emergencies ([58], [61]), the social community ([23]), and oncology ([44]; [63]). It is, therefore, a particularly suitable methodology for understanding users’ narratives in the field of health. As a quantitative and qualitative research tool, it consists of the in-depth analysis of the participant’s answers to the questionnaire and the subsequent denomination of Discursive Repertories (DRs) ([23]; [58]); these latter describe the ‘Congenital Heart Disease narrative’is shaped by respondents. Each DR is conceived as a particular way of generating the narrative and belongs to one of the following typologies (see also Appendix A):Generative Repertories (GR): they allow to produce a shift towards narratives that are different from those that have already become available;Stabilization Repertories (SR): they do not allow to produce a shift towards ‘other’ narratives from those that are already available, allowing them to keep the current one stable instead;Hybrid Repertories (HR): they can have both a generative and maintenance value, as they draw their value based on the class of the Repertories that they interact with in the narrative.

### 2.2. Data Collection and Analysis

To pursue the aim of our research we reached out to parents of children operated on for CHD. The recruitment took place through the collaboration with an association composed of doctors, educators, and volunteers, which offers support and assistance to families experiencing CHD. The association branches toward parents from different Italian regions—Veneto, Friuli-Venezia Giulia, Trentino-Alto Adige, Emilia-Romagna—and consists of about 150 members. Through the board of directors, the researchers sent a brief presentation of the research requesting availability to participate. Subsequently, the researchers also presented the research during assembly moments of the association. The written presentation of the research was sent to all members (about 150), while the presentation at meetings was made to about 80 people. Approximately 50 people volunteered to participate. Overall, about 33% of the association’s members participated in the research. We decided to close the data collection at 45 participants, when we noticed that data saturation had occurred, i.e., when participants’ responses began to repeat themselves.

In this preliminary research, we only referred to persons living on Italian territory. The only inclusion criterion for the study was the understanding of Italian language, while neither the children’s type of CHD nor the number of operations undergone was considered as an inclusion/exclusion criterion; thus, these latter information were not collected and used as variables.

Participants were administered a purpose-built questionnaire consisting of 4 open-ended questions, directly deriving from the research aims above mentioned (as shown in Table 1).

These two aims have been taken into account precisely because, as seen above, they are at the core of the establishment of a stigmatizing reality that makes the medical illness the sole pivot point around which the rest of the life can revolve.

Based on participants’ preference, the questionnaire was administered in written or oral form by an association’s operator. The association’s operators involved in administering the questionnaire had been trained in advance by the researchers; this allowed for them to remain maximally adherent to the questions, so as not to generate biases related participant’s formulation of their answers. Before starting to respond, each participant signed the informed consent to participate in the research. When administered in written form, questionnaires were compiled directly by the participants, in order to ensure that the answers’ text was not (even accidentally) rephrased or words altered. When administered in oral form, interviews were transcribed in their entirety, again without rephrasing sentences or changing terms. We received participants’ textual data already fully anonymized; being secondary use data, ethical review and approval were attained.

### 2.3. Scientific Data Validation

Credibility was achieved by specifying the researchers’ knowledge positioning, i.e., the epistemological and conceptual references described above. A second aspect concerned familiarity with the data, also given the collaboration with the association representing the participants. Furthermore, data were analyzed by all the researchers involved in the study, first individually and then as a group, with the aim of reaching a greater shared understanding of the textual data.

Regarding the transferability of results, the criteria for data classification were described and made explicit through the grid of DRs. This would allow for transferring the way in which data were collected. We have also provided specific data on participants, to precisely detail what the data refer to. In our case, the type of problem investigated (CHD) is very specific, as it concerns a specific congenital problem of participants’ children; therefore, data are specific and more transferable. A final element that strengthens transferability concerns the density of participants’ responses. Our specific method of analysis (MADIT) is based on a precise word-for-word collection of the participant’s text, thus, increasing accuracy.

The reliability of our work is given by the coherence between knowledge assumptions and data collection, and by the relationship between data and results. Within this process, the investigation protocols (outline) of the administered questionnaires were defined. The specificity of the analysis method also allowed for a more precise management of the difference between data and data interpretation. In our method, the collection part (Textual Analysis) and the discussion part (Intertextual Analysis) are distinct.

The researcher’s reflexivity part consisted of constantly monitoring the application of the method used. The critical elements detected in this process were reported before the conclusions in terms of research limitations. During the analysis, when it was noticed that the responses were not congruent with the purposes underlying the questions, we proceeded to contact the association’s operators (that in turn contacted the participant) or to discard the response.

## 3. Results

A total of 45 parents was recruited: 23 mothers (51.1%) and 22 fathers (48.9%), whose ages ranged from 31 to 67 (M = 47.86). Based on participants’ answers, we derived four different role configurations. For each one we will show the most frequent DRs used, as well as representative examples. We also highlight the main clinical and social repercussions grouped in clusters, again providing data on DRs used and text examples.

### 3.1. How Parents Configure Their Role

Starting from the description of the role of ‘parent of a child with CHD’, from the analysis of participants’ answer it was possible to extract four different role configurations, as shown in Table 2.

The most frequent DRs for each role configuration are shown in Figure 1.

#### 3.1.1. I’m Worried and I Will Protect Him/Her

The role configurations ‘I’m worried and I will protect him/her’ represents parents that describe and judge their role as parents as apprehensive, anxious, scared, overprotective or constantly on the alert, with answers like “*I am a very apprehensive person, we experienced very difficult moments*” (ID 5). They relate these feelings to their past experiences linked to their child condition—diagnosis, hospitalization, surgery, etc.—like ID 5 specifies: “*The birth and management of an ill child certainly accentuated my apprehension. Never would I have imagined myself standing outside the door of an intensive care unit with my child in danger of death*”. Other participants connote their emotion describing the current and future challenges they face—check-ups, medical relapses, daily activities, etc.—like ID 29: “Apprehensive and biased. I have a tendency to prevent my child from participating in activities that may overstrain him or I make sure that any ’heavy’ activities are carried out in complete health safety”. Using the Judgment DR—which morally labels attributes and/or personological features as inherent—to frame this picture (Figure 1A), both these features and the events they are associated with become established personal values for parents: this can be an issue as they turn potentially difficult to change when they negatively affect leading their life as parents’ of a child with CHD, also risking to become self-stigmatizing: “I describe myself as a mother afraid at first, then worried and now apprehensive about the heart disease and what we have experienced in the past” (ID 14).

#### 3.1.2. I’m a Hero!

Parents’ role configuration ‘I’m a hero!’ accounts for participants that describe themselves as brave, mature, aware, like ID 20: “I feel like a strong mother; I found myself very strong, which I didn’t think I was” (ID 20). Epitome of these features is the ‘heroic’ parent, able to overcome the adverse events and the difficulties linked to his/her child pathology and prepared for whatever may come next: “I feel like a strong and important parent, with an armor that allows me to face everything” (ID 10). This heroic figure is not only recognized by parents themselves, but also by what they imagine others think of them, like stated by ID 27: “When you tell someone about your experience, you are seen as a hero, as a great person who was able to handle a situation that for many might seem unbearable”. Also for these participants, the ‘strong’ characteristic of parents becomes an intrinsic personal value through the Judgment DR (Figure 1B), risking to not leave space for other features or feelings: “The various situations, both positive and negative, have definitely strengthened me in character to be able to overcome them” (ID 30). Being a “hero”, they cannot yield or fail, but what happens if they do?

#### 3.1.3. I’m Normal and Untroubled

To this role configuration belong participants that describe themselves just as ‘parents’: “I am a parent, a normal parent, an ordinary parent” (ID 19). While they mostly use the judgment narratives (JM repertory) to do this, differently from the two previous profiles they support and motivate this connotation through the evaluation narratives (EU repertory, which expresses the criteria used to state the answer given, even if these are personal and not necessarily agreeable) (Figure 1C). In particular, they refer to the child being the only one or the only one with a medical condition, thus, not having something to compare, like ID 37: “In our case we have no comparison, I have no other children with illness, so a normal parent”. Alternatively, they refer to the situation being normal and being serene about it, like ID 33: “I describe myself as quite serene, I think thanks to several factors. 1) We learned during the pregnancy of the child’s heart disease, so we had time to prepare ourselves psychologically for what could have happened”. Thanks to the use of evaluation narratives, “just being a regular parent” results as a flexible element in participants’ narrative, making them more adaptive to uncertainties and unexpected events: “For me it’s normal, it’s not that because he had had the surgery I treated him differently. I have another older son and I’ve behaved the same way as a parent” (ID 42).

#### 3.1.4. I’m Unfortunate

The fourth role configuration (‘I’m unfortunate’) represent parents that are depicted as “victims” of an unfortunate adverse event, as exemplified by ID 21: “At the beginning I felt unlucky because of the situation, you know, because of the fear […] but…yes, it was unfortunate and it definitely wasn’t easy”. This is particularly true in relation to what participants think others describe them, as reported by ID 1 “I am afraid that others think of us, as parents of cardiopathic children, as ‘poor guys, look what happened to them’” and ID 17 “Others might describe the parent as an unfortunate person, with a load of problems and responsibilities”. In some answers, this is grounded by parents’ specific evaluations; however, in the majority of them, being ‘unlucky’ and a subject of an unfortunate event becomes a moral attribute attached to parents in a judgmental manner, thus, making it a firm feature of that parent personality/character (Figure 1D): “They think of me as an unfortunate parent; I feel it on me, they often say ‘how do you do it, poor, little one’” (ID 43).

### 3.2. The Clinical and Social Repercussions

We identified 4 main repercussions’ clusters, as shown in Table 3.

Figure 2 reports the most frequent DRs for each repercussions’ cluster.

#### 3.2.1. State of Ordeal

For the cluster ‘State of ordeal’, from participants’ answers it appears that CHD definitely brings fear, worries, and anxiety on parents, leading to further repercussions (such as increased and constant attention) both for the parents and the child. This is clearly exemplified by ID 3: “My son’s condition certainly has consequences on the way I relate to him and his safety: compared to my other son, for instance, I realize that I am more concerned for him, especially when he has to face experiences that are new to him and to us”. This text highlights the use of a certifying and judgmental narrative, where the parent does not foresee alternatives (Figure 2A).

In a similar way are described concerns directly linked to the child’s health condition or to medical-related issues, as ID 1 and 31 say: “Check-ups, examinations, etc. always cause preoccupation, even if they are limited to their duration or to waiting for the results”; “There have been implications for me. There have been times, at the diagnosis of heart disease for example, when you think that it can’t be happening to you. The worries are so intense that you don’t know where to start to deal with them and you honestly don’t even know if you’re going to cope”. The second answer, in particular, establishes (CR repertory) that the parent reality is and can only be that one (Figure 2A); based on that, the parent’s coherence cannot contemplate possible future scenarios other than ‘not knowing if you will cope’.

Otherwise, some parents mention social and work-related concerns, as ID 11 does: “We can talk about stress, worries, periods of absence from work. How will society deal with my daughter? I know that I will have to take her by the hand for a long time”. Other parents, instead, show a more widespread concern, such as ID 5: “Everything, even the most trivial, worries me because I think it may have implications for the heart”. Although here the worry outlined is broader, the respondent employs a more generic and flexible narrative (EU repertory), where the reasons for concern are less impactful and more open to change.

#### 3.2.2. State of Alert and Overprotection

The cluster ‘State of alert and overprotection’ accounts for parents becoming highly protective as a result of their child’s diagnosis and medical condition: they judge themselves as too careful and cautious, establishing these attributes as psychological trait and moral characteristic (JM and CR repertories) (Figure 2B). The answer from ID 2 states the following: “There is a tendency for us parents of children with heart disease to be too protective of them”. Stating that there is a generalized ‘tendency’ to be overprotective serves only to confirm this type of narrative, leading to the risk of assuming a stigmatizing frame. In addition to this, ID 8 says: “I became more careful and alert to my son’s needs and consequently also to the needs of my second son”. This text neatly represent that such judgmental narrative can spread, becoming a generalized tendency to be overprotective not only towards the ill child, but also to their siblings. By continuing in this direction, it could also affect other aspects of their personal and family life.

#### 3.2.3. Limitation of Life Experiences

For the cluster ‘Limitation of life experiences’, parents’ outline more concrete repercussions on their lives, such as depriving themselves from certain experiences or activities due to the child’s illness: “Turns out that we have to deprive ourselves of some experience because of the child’s condition” (ID 7). Some of these limitations are clearly described by participants, as exemplified by ID 35: “the implications were there, limitations and additional tasks that affected all family life…less time, less space to do other personal things …”. More frequently, instead, parents connect activity restrictions to other repercussions, using the repertory of Implications (Figure 2C); this type of narrative causally links a given element with others potentially stemming from it. Two examples are brought by ID 36 and 29; in the former is the perception of the child’s medical condition that produces psychological issues, while in the latter it is the parent’s psychological state that leads to restrictions: “I kind of have this thought, this psychological condition, in thinking how this heart disease limits her”; “Sometimes my excessive concern for his health causes me to limit some activities that we could do as a family”.

#### 3.2.4. Personal Identity Changes

The cluster ‘Personal identity changes’ represents parents stating that they experienced a distinct personological change in their character due to the situation they had to deal with. In their answers they firmly declare this relationship between the child’s condition and their character change (Figure 2D), as it can be seen from the following answers: “My child’s illness changed me character-wise all round. It forced me to break out of the shell of stereotypes I was in on a personal, relational, work level, making me a different person” (ID 12); “The discovery of the malformation, the visits, the hospitalization, the day of the surgery are situations that left their mark on me, changing my personality. I recognize that I’ve become much more apprehensive than I was before” (ID 4). These two answers also highlight that the parents’ personality transformation went in different directions: ID 4 became more scared and preoccupied, while ID 12 became more brave and self-aware. A third direction is outlined by ID 26, who affirms through the Certify Reality DR: “Since the discovery of heart disease I empathize much more with human qualities related to suffering. I also have less tolerance of the arrogance of people and poor moral profiles”.

## 4. Discussion

Drawing from the results outlined, we can state that the role of parent of a child with CHD revolves tightly around the features represented by the four role configurations identified, which become generalized personal tendencies and narratives through the major use of Stabilization DRs.

Starting with ‘I’m worried and I will protect him/her’, it is a parent made anxious, apprehensive, on alert against potential problems and threats ([53]; [66]). This lead the parent to be protective toward his/her child, fluctuating between being caring and at times controlling. This aligns with the findings from ([1]; [53]), for which feelings of overprotection and control came as a consequence of anxiety, fear, and helplessness. By judging their role in this firm and ‘locked’ way (Judgment and Opinion DRs), such connotation risks becoming a totalizing characterization, where the parent cannot be serene and limits engagement possibilities (e.g., experiences, social activities, etc.) both for the child and the entire family ([1]). The Judgment DR, being used more than twice as often as other DRs, acts as a ‘black hole’ that attracts other narrative possibilities back to itself. Indeed, in being totalizing and, thus, tending to ‘personify’ the characteristics, it is possible that parents reject, without a proper criteria or motivation, any unexpected or non-experienced events, even if positive. Moreover, they could ask for the support of relatives and professionals in dealing with their feelings of worry and fear; however, these help requests risk remaining general and generic, not being linked with a specific difficulty but deriving instead from a narrative full of dangers and worries.

In a similar way, also the role configuration ‘I’m a hero!’ incurs the same risk of a totalizing typification. The child’s CHD-related events made the parent strong, aware, and, thus, in full control of what happened. Parents use these adjectives as moral judgments, and these connotations become the sole elements characterizing the role, to the point of turning into a hero. However, this reduces the possibility of playing other roles both within and outside the family context, necessarily having to be the strong figure for the child, the family, and themselves. Again, the ample use of the Judgment DR is what reduces such possibilities. A parent from the study of Franklin et al. stated “Several times I have had to hold it all inside and stay strong for my wife and child”, corroborating the presence of an ‘imposed’ social expectation (both internal and external) to ‘stay strong’ ([16]). If the events fall within the spectrum of those already experienced and/or contemplated by the parent, assuming the role of hero can certainly be a valuable resource for the entire family ([4]). However, if non-experienced or unanticipated difficulties arise, parents may struggle to recognize that they are unable to handle the situation alone and that others can be effective supporting roles. In this way the burden of care could remain ‘an own experience’ placed solely on the parent (Lawoko & Soares, 2002; Tak & McCubbin, 2002; Vainberg et al., 2019). This could lead in turn to the difficulty to ask for help from others, whether these are relatives or professionals.

Moving to ‘I’m normal and untroubled’, the role is depicted as not ‘rattled’ and by the child’s pathology and the events related to it, in contrast to what is often reported by field literature ([6]; [18]; [41]). In fact, whereas some parents report a sense of non-normality due to the child’s condition, others instead try to adapt to and ‘normalize’ the life of a son/daughter with CHD ([9]; [17]). Thus, the parent narrates themselves as normal, trying to make their child feel normal as well. These parents appear to be serene and hopeful for the future. Different from the other three role configurations, being ‘untroubled’ is grounded by specific motivations and criteria (expressed through the Evaluation DR); this reduces the ‘restricting’ effect of the Judgment DR and helps parents clearly understand the reasons behind their ‘serenity’ and, thus, to be aware of the situation they are living in. However, where such criteria are missing and only the Judgment remains, the narrative opens up to the risk that an untroubled parent does not recognize potential difficulties or issues for their own life or the child’s life; the parent may let things slide and not act promptly and consciously to resolve them, precisely because they do not ‘weigh’ them adequately.

Lastly, the role configuration ‘I’m unfortunate’ is a parent to whom an unexpected and—indeed—unfortunate event has happened; it is related to randomness. The connotation of ‘unfortunate’ is sometimes linked to the events per se ([53])—as also reported by a participant ([31]), who stated “Such an unfortunate affliction only came to me, and I felt resentment towards the world”—while other times it is used as an adjective that negatively characterize the role. This occurs especially when parents imagine how others see them; they think others have pity or sympathy for them, thus leading to assume a stigmatized role. In addition, if the element of ‘unluckiness’ is used as the only relevant element of the narrative (and not one of several facets of it) it could lead the parents to state that “there is nothing to be done”: this framework, resulting from the extended use of the Judgement DR, weakens the management of criticalities (e.g., saying “sadly, this also happened to us”), thus, reinforcing stigmatizing rhetorics and attitudes.

Moving to the impact of CHD’s psychological and social consequences on parents’ lives, we observed that almost all repercussion clusters are depicted as firm and permanent elements, judged as an integral part of that person’s existence. Parents see these repercussions as the only inevitable outcome of the events they have experienced (Certify Reality DR), and struggle to foreshadow different alternative scenarios. This is particularly true for the cluster ‘State of concern’: fears, worries, apprehension are described as directly linked to their child’s pathology and to the events they had to experience, and for parents it could not be different from that, as also corroborated by ([64]), in which parents’ reported emotional distress, fear, uncertainty, and helplessness. Framing the psychological repercussions in such monolithic way risks in the first place the generating of typification and stigmatization. In second place, it also raises the possibility that the repercussions spread, and their negative impact transfers to other life aspects, as precisely see in the cluster ‘State of alert and overprotection’. Only the cluster ‘Limitation of life experiences’ appears to be less monolithic, as it described without strong moral or personal connotation, but instead with more understandable links and argumentations. This is also reflected by other research. On one hand, restrictions on social and physical activities resulting from the CHD are highlighted both for children and parents ([41]; [49]) but at the same time the condition can be configured as a ‘driver’ to make them: “It has caused us to do things in our lives at a different point than we would have otherwise. […] there are things we have done in our lives, because we wanted to make sure that they had the opportunity to do those things when there were no health issues” ([9]). This could favor health professionals to transform the current framework towards scenarios in which the implications do not exhaust the narrative possibilities, reducing in turn the associated stigmatization and improving parents’ health and hopefulness towards the future ([33]).

## 5. Conclusions

Our research highlighted how parents’ narratives about their role and the impact of the psychosocial repercussions of their child’s condition on their life are shaped in a pervasive and totalizing manner. Parents do not consider nor anticipate different alternatives to the role they assume, independently of which this is among the four we identified. In addition, they depict the psychological and clinical repercussion (or the absence of them) as inevitable facts, firmly established in their life since the discovery of the child’s heart condition. These data and observations, coming directly from parents’ narratives, can be leveraged by health professionals in favor of both parents and children with CHD. Indeed, knowing which discursive modalities parents employ to describe their role, experiences, and issues is the first step to initiate an intervention aimed to change them—especially if they belong only to the Stabilization typology—or reinforce them—if Generative. In turn, this would allow one to increase the effectiveness of the support they could provide toward the other parent, the child, and the whole family.

In light of the specific outcomes of our research, we think that dedicated clinical and social interventions are needed to reduce the risk of both self-stigmatization and hetero-stigmatization, as well as exacerbating the already impactful difficulties, challenges, and negative feelings. These interventions should be tailored to the particular profile of the parent and the discursive modalities they use in recounting their experience in the role of parent of a child with CHD. As an example, for the profile ‘Parent as a Protective and Worried person’, health professionals like psychologists and case-managers should try to transform the judgments and connotation in motivated statements, making parents describe more in depth how they feel and why. This will make them adopt more hybrid and generative modalities, allowing them to conceive their concerns and fears in a more balanced way, without exhausting the narrative only around these in a potentially stigmatizing way. Referring to the profile ‘Parent as a hero’, field professionals could frame hypothetical scenarios that could occur from a certain repercussion and that the parent did not think of; this should promote more evaluations, considerations, and anticipation by parents, making them no longer ‘just’ strong heroes but skilled and prepared managers of their life and their family’s life.

Thanks to the methodological work performed, we were able to further and better qualify the narrative structures of parents of children with CHD, related to their role and the psychological and health repercussions of their child’s condition. We believe that leveraging directly these parents’ narratives is crucial and could lead to new and further insights on the processes they use to conduct their everyday lives in conjunction with the management of their child’s pathology. These insights, briefly outlined through the exemplary intervention outlined above, should allow the boosting of the effectiveness of actions aimed at promoting psychological health and wellbeing for these parents ([62]). However, it is clear to us that it does not come without a strong effort. Indeed, in order to make this step the training of the healthcare professionals on the interactive management of patients and caregivers is necessary ([59]); working on skills, as well as on the observation of interactions and the anticipation of the consequences of the communication strategies used will empower professionals to choose a more effective way for the design and provision of psychological interventions.

## Figures and Tables

**Figure 1 behavsci-15-00269-f001:**
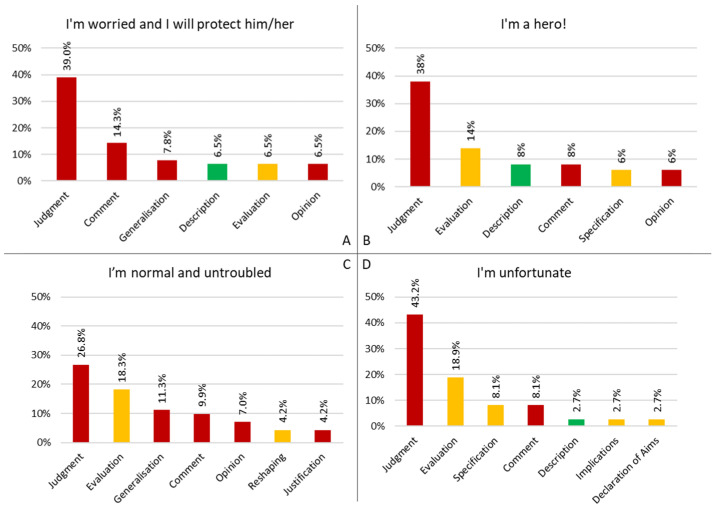
Main DRs and related frequency for each role configuration. (**A**) role configuration “I’m worried and I will protect him/her; (**B**) “I’m a hero!”; (**C**) “I’m normal and untroubled”; (**D**) “I’m unfortunate”.

**Figure 2 behavsci-15-00269-f002:**
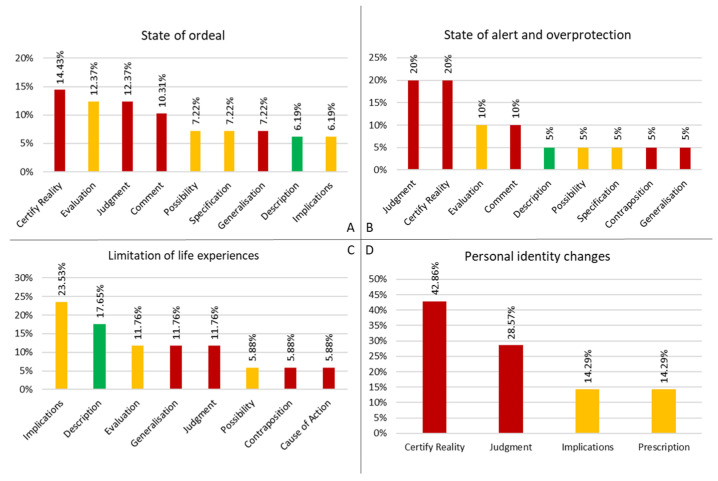
Main DRs and related frequency for each repercussions’ cluster. (**A**) repercussion cluster “State of ordeal”; (**B**) “State of alert and overprotection”; (**C**) “Limitation of life experiences”; (**D**) “Personal identity changes”.

**Table 1 behavsci-15-00269-t001:** Research aims and questions, with examples of answer and DR denomination.

Research Aims	Questions	Example of Answer	DR
Describing the role of ‘parent of a child with CHD’	How would you describe yourself as a parent of a child with congenital heart disease?	*I think I learned so much from suffering and fear. But I definitely would have preferred to remain more ignorant, more superficial but more carefree.*	Contraposition (SR)
	How might people who have not experienced congenital heart disease describe a parent of a child with such a condition?	*We are usually told that we are strong,* *but I think others cannot totally understand our fears and emotions.*	Judgment (SR) Specification (HR)
Describing the impact of CHD’s psychosocial repercussions on parents’ lives	As a parent of a child with congenital heart disease, how would you describe the repercussions your child’s heart condition has had on you?	*I think there is more concern about what happens to him, in the sense that even every little illness or injury leads to “other” reflections and what might happen to him in his condition.*	Evaluation (HR)
	How might people who have not experienced congenital heart disease describe the repercussions of the child’s heart condition on his/her parents?	*Apprehension, fear, anxiety.*	Certify Reality (SR)

**Table 2 behavsci-15-00269-t002:** Role configurations identified from participants’ answers.

Role Configurations
I’m worried and I will protect him/her
I’m a hero!
I’m normal and untroubled
I’m unfortunate

**Table 3 behavsci-15-00269-t003:** Repercussions’ clusters identified from participants’ answers.

Repercussions’ Clusters
State of ordeal
State of alert and overprotection
Limitation of life experiences
Personal identity changes

## Data Availability

The untranslated original raw data presented in the study are openly available at [https://docs.google.com/spreadsheets/d/1TvVdlX0aSQUaMsJ_FTAXvybFeJBkGF6M/edit?gid=27510299#gid=27510299] (accessed on 2 February 2024).

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
