# Peer review of "Parents of Children with Congenital Heart Disease (CHD): A Narrative Study of the Social and Clinical Impact of CHD Diagnosis on Their Role and Health"

_behavsci, 2025, doi:10.3390/bs15030269_

Round 1

Reviewer 1 Report

Comments and Suggestions for Authors

The paper titled “Parents of children with congenital heart disease: a narrative 2 study of the social and clinical impact of CHD diagnosis on 3 their role and health” deals with an interesting and sensitive topic with a potential impact on the experiences of parents’ and carers’ of children with a CHD. Overall, the article is well-written, and the methodology used seemed relevant and appropriate. I would recommend some modifications to improve the article.

1.     There lacks a clear argument in the introduction section as to why you are conducting this secondary analysis of data. How will this specific type of data analysis add value to parents/carers and the delivery of healthcare in future?  

2.     Figures 1-4 and 5-8 might be more useful organised as 2x2 matrices to allow the comparison of findings across the clusters more clearly.

3.     The discussion mainly just summarises the results section. How do your findings compare to the wider literature? Does this analysis add anything further to previous qualitative studies with parents of children with CHD?

4.     I struggled to understand the significance and impact of the findings. Could you further strengthen the conclusion section, and make it clearer what the practical or theoretical implications of the findings might be?

Author Response

First of all, we would like to thank the reviewer for the appreciation and the precious suggestion provided. Here follows the point-by-point respons to the comments:

1. There lacks a clear argument in the introduction section as to why you are conducting this secondary analysis of data. How will this specific type of data analysis add value to parents/carers and the delivery of healthcare in future? 

The research origins from a partnership between the University of Padua and the Association “Un Cuore Un Mondo Padova Onlus”. In order to improve the quality of the Association’s interventions and services, it was necessary to further and better qualify the parents’ experience. To do so, we adopted one of the most relevant mean available, that is textual analysis. With the Association we shared the opportunity to help children with CHD by trying to enhance the effectiveness of their parents’ support. The idea was to improve the indirect care toward people with CHD. We briefly added the above at the end of the Introduction section.

2. Figures 1-4 and 5-8 might be more useful organised as 2x2 matrices to allow the comparison of findings across the clusters more clearly.

Done; we have now changed the figures as suggested and adjusted their references in the text accordingly.

3. The discussion mainly just summarises the results section. How do your findings compare to the wider literature? Does this analysis add anything further to previous qualitative studies with parents of children with CHD?

We thank the reviewer for this input. As suggested, we added some more literature references to compare outputs in the Discussion section. We also exploited a little more the data analysis value of DRs, expressing in more detail their effect when used in parents’ narratives.

4. I struggled to understand the significance and impact of the findings. Could you further strengthen the conclusion section, and make it clearer what the practical or theoretical implications of the findings might be?

We understand this issue, and tried to address it by adding some more specifics on the pragmatic use of our observations in the first part of the Conclusions section.

Reviewer 2 Report

Comments and Suggestions for Authors

Thank you for allowing me to review this manuscript describing the authors’ work exploring experiences and perceptions of those experiences of having children with congenital heart defects.  In addition to needing substantial editing for language, I have the following comments and suggestions for the authors.

Introduction

The authors give a concise summary of the literature around stressors faced by parents of children with congenital heart disease.  This section is appropriately cited and referenced.  A bit more detail in some areas would be helpful. For example, social stereotyping, isolation, and stigmatization is mentioned without further definition or description of context, which would help to establish the significance of the study better. 

Methods

The methods section needs further detail of the recruitment process.  The authors state (line 117) “a sample of 45 parents of children operated for CHD has been recruited.”  This statement should be under the Results section.  In Methods, please describe the way in which these parents will be approached and asked to participate.  What was the recruitment goal?  This information is somewhat addressed in the paragraph starting with line 131.  Expound on the ethical review of the study. 

The framework and methodology of the study pose the greatest issue for me with this manuscript.  Over a third of the 69 citations for this manuscript are self-citations of these authors, including all of the citations for the framework, methodology, and examples of how the framework and methodology have been used previously.  Are there examples of others using this methodology outside this group?  This could be a key issue with the validity of this article.

Results

Please provide a table of the participant characteristics.  This should include the level of CHD classification.  Those with children who have small ventricular septal defects will have very different experiences than parents of children with hypoplastic left heart syndrome. 

How many parents were recruited to obtain the sample size of 45? 

The last line of the Results section has a citation that needs to be formatted.

Discussion and Conclusions

Although this was not a study to test the methodology MADIT, one of the authors’ conclusions was that “MADIT could lead to new and further insights on the processes they use to conduct their everyday lives in conjunction with the management of their child’s pathology (being that a CHD or other chronic conditions).”   It may be a stretch that the results of this study generalize to a population of other chronic conditions or that the methodology is supported for use outside of the results of this study.  I am concerned again that the citations used in this paragraph of the conclusions section are those of the authors. 

Comments on the Quality of English Language

The manuscript needs overall editing for English language.

Author Response

First of all, we would like to thank the reviewer for the precious suggestion provided. Here follows the point-by-point respons to the comments:

Introduction

The authors give a concise summary of the literature around stressors faced by parents of children with congenital heart disease.  This section is appropriately cited and referenced.  A bit more detail in some areas would be helpful. For example, social stereotyping, isolation, and stigmatization is mentioned without further definition or description of context, which would help to establish the significance of the study better. 

We thank the reviewer for the comment and agree with the assessment. Following the suggestion, we added some more details and references regarding the areas mentioned.

Methods

The methods section needs further detail of the recruitment process.  The authors state (line 117) “a sample of 45 parents of children operated for CHD has been recruited.”  This statement should be under the Results section.  In Methods, please describe the way in which these parents will be approached and asked to participate.  What was the recruitment goal?  This information is somewhat addressed in the paragraph starting with line 131.  Expound on the ethical review of the study. 

We added some more details regarding the recruitment process and how parents have been approached. We moved the sentence with the total number of participants in the Results section as suggested. We also added some more information about the ethical review at the end of the methods section (synthesizing what was already mentioned in the devoted statement at the end of the paper).

The framework and methodology of the study pose the greatest issue for me with this manuscript.  Over a third of the 69 citations for this manuscript are self-citations of these authors, including all of the citations for the framework, methodology, and examples of how the framework and methodology have been used previously.  Are there examples of others using this methodology outside this group?  This could be a key issue with the validity of this article. 

We thank the reviewer for highlighting this issue. First of all, we reduced the number of self-citations, removing 8 of them. We look forward to any further indication on this, if the current amount of self-citation is still considered too high. Regarding the specifics of MADIT, it is a methodology mainly developed in the University of Padova, whilst applied in several international researches and project. Many of these authors don’t currently belong to the same research group. However, pertaining to the methodologies mainly focused on qualitative research, and specifically on discourse analysis, we added some more specifics on that in the paper.

Results

Please provide a table of the participant characteristics.  This should include the level of CHD classification.  Those with children who have small ventricular septal defects will have very different experiences than parents of children with hypoplastic left heart syndrome. 

Considering the relatively small size of our sample, we did not use participants’ characteristics (e.g. age, gender, etc.) as research variables. This also because, with this preliminary research, the association was mainly interested in having aggregated data on that overall type of population. That is also why the only inclusion criteria was the understanding of the Italian language and the reason why the child’s specific CHD classification was not collected from parents.

How many parents were recruited to obtain the sample size of 45? 

Being secondary use data, the 45 parents participating in the research is also the total number of people recruited. 

The last line of the Results section has a citation that needs to be formatted.

Adjusted.

Discussion and Conclusions

Although this was not a study to test the methodology MADIT, one of the authors’ conclusions was that “MADIT could lead to new and further insights on the processes they use to conduct their everyday lives in conjunction with the management of their child’s pathology (being that a CHD or other chronic conditions).”   It may be a stretch that the results of this study generalize to a population of other chronic conditions or that the methodology is supported for use outside of the results of this study.  I am concerned again that the citations used in this paragraph of the conclusions section are those of the authors.

We thank the reviewer for the comment. While MADIT methodology has been used in other contexts (e.g. oncology, ADHD, etc.), we agree that the specific results of this study do not necessarily apply to other chronic conditions without dedicated research. Thus, we proceeded to change that part and made some other smaller changes, as well as reducing the number of self-citations (as mentioned above).

Round 2

Reviewer 1 Report

Comments and Suggestions for Authors

The authors have successfully addressed my previous comments.

Author Response

We thanks the reviewer for the feedback and the previous comments, that helped to improve the manuscript.